# Influencing factors and configuration pathways of fundraising effectiveness in charitable foundations: An FSQCA analysis of 54 cases in China

Zhe Zhu[1,2], Shan Huang[3], Wanqiang Xu[4,5], Heran Zou[6]*

1 College of Management, Wuhan Institute of Technology, Wuhan, China, 2 Enterprise and Environment Coordinated Development Research Center of Hubei Province, Wuhan, China, 3 College of Management, Wuhan Institute of Technology, Wuhan, China, 4 School of Public Administration, Huazhong University of Science and Technology, Wuhan, China, 5 School of Emergency Management, Henan Polytechnic University, Jiaozuo, China, 6 College of Management, Wuhan Institute of Technology, Wuhan, China

* 22406010039@stu.wit.edu.cn

## Abstract

Charitable foundations play an important role in assisting vulnerable groups, narrowing the wealth gap, and addressing challenges such as urban-rural and regional development imbalances. Unlike other nonprofit organizations, fundraising effectiveness serves as a vital lifeline for the sustainable and stable development of charitable foundations. This study analyzed 54 Chinese charitable foundations and employed the fsQCA method to explore the factors influencing their fundraising effectiveness and their configuration effects. Research has found that: (1) digital transparency infrastructure and program activity intensity are key conditions for improving the fundraising effectiveness of charitable foundations; the level of professional management and political connections exert important impacts on enhancing the fundraising effectiveness of foundations, serving as important influencing conditions for the improvement of fundraising effectiveness. (2) There are two core paths for charitable foundations to achieve high fundraising benefits: one is a combination of strong program activity, robust digital transparency, and high political connections; the other is a combination of strong program activity, robust digital transparency, and high professional management. This study enriches and expands the theoretical connotation and research paradigm of charitable fundraising research and provides empirical evidence and policy guidance for promoting the sustainable and healthy development of charitable foundations.

## 1. Introduction

As of 2024, the global charitable foundation market is estimated to be around $1.625 billion and is projected to reach $2.291 billion by 2031. Charitable foundations play

**Data availability statement:** All data underlying the findings described in this manuscript are fully available from a public database. Specifically, the annual reports and foundational information related to environmental protection social organizations and charitable foundations in Hubei Province were collected from the China Social Organization Public Service Platform, hosted by the Ministry of Civil Affairs of the People's Republic of China (URL: https://xxgs.chinanpo.mca.gov.cn/). All data on this platform are public information and can be accessed and downloaded without restriction by any user. Therefore, the data supporting the findings of this study are fully publicly available.

**Funding:** This study was supported by the National Natural Science Foundation of China (Grant Number: 72004172) and the Humanities and Social Sciences Platform Innovation Fund of Wuhan Institute of Technology (Grant Number: 2025RWSKPTCXJJ04) awarded to Z.Z. The funders are the National Natural Science Foundation of China and the Enterprise and Environmental Coordinated Development Research Center, a provincial-level platform at Wuhan Institute of Technology. The funders' websites are: https://www.nsfc.gov.cn/ and https://www.wit.edu.cn/. The funders had no role in study design, data collection and analysis, decision to publish, or preparation of the manuscript.

**Competing interests:** The author has declared that there is no conflict of interest.

an irreplaceable role in narrowing the wealth gap, supporting vulnerable groups, and participating in social governance [1]. Through the third distribution of social resources, they effectively compensate for the limitations of government and market functions, serving as a key force in maintaining social stability and promoting sustainable development. By the end of 2024, the number of charitable foundations in China had reached 13,000, with total net assets nearing ¥200 billion. A total of 1,544 charitable trusts had been established nationwide, with a contract value of ¥6.333 billion. Under the guidance of national strategies, charitable foundations have been endowed with significant historical missions. The Chinese government aims to promote the development of charitable foundations to guide the third distribution, reduce wealth disparities, address imbalances in urban-rural and regional development, and advance the goal of "common prosperity" [2]. However, the survival and sustainable development of organizations depend on their ability to acquire resources [3]. Unlike other nonprofit organizations, charitable foundations rely heavily on social donations rather than service fees or government subsidies. Thus, fundraising effectiveness is a core factor that influences their survival and development.

Currently, although the role of charitable foundations is increasingly prominent and their funding scale continues to grow, the development of charitable foundations in China has shown a significant "Matthew effect" in their development process [4]. As a systemic bias in the development of charitable foundations, the Matthew effect leads to the concentration of resources in top-tier foundations, while grassroots foundations fall into fundraising difficulties. This unbalanced development is inherently linked to the "causal asymmetry between high and low fundraising efficiency" identified in this study: the core advantages of top-tier foundations (intensity of program activities, digital transparency infrastructure) further strengthen their resource acquisition capacity, whereas the shortcomings of grassroots foundations (low intensity of program activities, inadequate professional management) are amplified by the Matthew effect, ultimately forming the causal asymmetry characterized by convergent high-efficiency configurations and divergent low-efficiency configurations. The vast majority of organizations are deeply trapped in fundraising challenges, with many struggling with low fundraising effectiveness and unstable funding sources, which severely constrain their organizational growth and social functions [5,6]. In practice, charitable foundations often remain dependent on traditional paths, revealing deep-seated problems such as "emphasizing scale expansion over efficiency," "prioritizing resource acquisition over brand maintenance," and "weak public trust." Against this backdrop, this study analyzes 54 national charitable foundations in China, employing the fuzzy-set Qualitative Comparative Analysis (fsQCA) method to break through the limitations of traditional linear analysis and explore the factors and configuration effects that influence fundraising effectiveness. The study addresses two core questions: (1) Which conditions are more critical for improving the fundraising effectiveness of charitable foundations? (2) What equifinal configurations promote high fundraising effectiveness in charitable foundations?

Theoretically, this study contributes by examining the complex causal mechanisms that affect the fundraising effectiveness of charitable foundations, identifying key



influencing factors from organizational, behavioral, and institutional perspectives, and enriching the application of resource dependence theory and institutional theory in nonprofit organization research. Moreover, it advances charitable fundraising research from a "factor identification" paradigm to a "configuration interaction" perspective. Since different factors do not independently influence fundraising effectiveness, their linkage matching also impact outcomes to varying degrees. Adopting a "configuration perspective" deepens our understanding of the complex mechanisms underlying charitable fundraising. At the practical level, the study provides a clear roadmap for the development of charitable foundations. It helps them identify core elements and effective strategy combinations for improving fundraising effectiveness, improve their funding acquisition pathways, increase funding acquisition efficiency, and help foundations avoid falling into low-performance fundraising traps. This optimizes organizational resource allocation and promotes sustainable organizational development. On the other hand, the research offers empirical evidence for relevant government agencies to formulate more targeted growth incentives and oversight policies for charitable organizations, fostering a healthy, efficient, and transparent philanthropic ecosystem.

## 2. Literature review

Existing research on fundraising in charitable foundations can be broadly categorized into three interrelated levels: macro, meso, and micro—namely, institutional environment, organizational capacity, and donor trust. This review will follow this framework to synthesize existing literature and identify theoretical gaps for this study.

### 2.1 Institutional environment

Organizations cannot exist in isolation from their institutional environments. For charitable foundations, public trust is a prerequisite for fundraising and sustainable development, and maintaining organizational legitimacy is key to sustaining trust [7,8]. Suchman (1995) defines legitimacy as the perception or assumption that the actions of an entity are generally regarded as desirable, appropriate, or proper by the norms, values, beliefs, and definitions within its social system [9]. In the Chinese context, national laws and policies constitute the core of the institutional environment. The enactment of the Charity Law provides a legal foundation for fundraising activities of nonprofit organizations. Institutional theory suggests that organizations must also cultivate deeper cognitive and normative legitimacy through mechanisms like imitating successful models or adopting professional norms to gain stakeholder trust [10]. At the same time, the rise of digital technology, particularly the "internet Plus" fundraising model, has profoundly reshaped the ecosystem for charitable foundations. This model transcends geographical limitations and has become a major channel for public participation in philanthropy, breaking the existing situation where charitable organizations are confined by administrative management systems and restricted by administrative boundaries [11]. The fundraising eligibility and legitimacy of nonprofit organizations depend on organizational formality and the legitimacy of fundraising activities, serving as an "entry ticket" for fundraising activities [12]. While the Charity Law has brought order to previously online fundraising which generated by the development of science and technology, challenges such as immature application of internet technology, user privacy breaches, low platform credibility, and inconsistent fundraising management measures persist, hindering the development of charitable organizations [13]. Wang Dan (2014) believed that it was necessary to optimize the top-down qualification confirmation system with the network platform fundraiser as the core and strengthen inside-out regulatory system of the network fundraiser [14]. Internationally, crowdfunding platforms have become key channels for charities to raise funds for social sustainable development [15]. Although this model is very popular, it also brings new challenges. many campaigns fail to meet their targets within deadlines [16]. A number of studies highlight platform credibility [17], information transparency [18], and technological maturity [19] as critical factors influencing online fundraising success. Poor platform governance can erode trust and diminish donor willingness. Therefore, technology itself is neutral; its ultimate impact depends on how it is applied within a sound institutional framework to establish and maintain trust.

## 2.2 Organizational capacity

In a competitive resource environment, a foundation's internal governance and strategic capabilities are crucial. Resource dependence theory emphasizes that no organization is self-sufficient; all must interact with their external environment to survive [20]. From an internal governance perspective, with the maturation of online fundraising, traditional reliance on government resources has shifted to a dual dependence on both government and societal resources, forming the primary state of resource acquisition for charitable organizations, which means that the way of resource acquisition of charitable organizations will not only be limited by the system, but also affected by the resources of other social [21]. This dual dependence compels foundations to develop diversified strategies for managing resource channels. Xu et al. (2017) through survey analysis of Shanghai's online charitable fundraising practices, found that low enthusiasm in industry management hindered the development of public welfare in Shanghai, and innovating industry management systems could improve this situation [22]. Wu and Chen (2021) argue that the organization providing official public services or products has the legal support and the qualification to raise resources publicly [23]. Charitable organizations must effectively bridge institutional and societal resource acquisition to achieve dual fundraising benefits. In this context, program activity intensity and professional management emerge as key organizational capabilities. Sufficient program activities act as a signal to effectively reduce information asymmetry faced by donors, strengthen organizational brand building, and enhance their trust and willingness to donate [24]. Waters (2011) found that effective relationship management with nonprofit organizations significantly increases donor loyalty and willingness to give [25]. Additionally, establishing political connections with key resource holders, such as the government, has been proved to be an effective way in acquiring critical resources and legitimacy, particularly in environments with high institutional uncertainty [26].

## 2.3 Donor trust

The realization of fundraising effectiveness relies on the decision-making behavior of individual donors at the micro level. Existing research explores the psychological mechanisms that drives individual donations, supplementing traditional economic models with insights from behavioral economics and psychology. Beyond altruism, factors such as organizational influence, trust, social norms, and emotional arousal are powerful drivers [6]. In the online environment, these psychological factors are amplified. Whether online donation platforms provide clear, traceable feedback on how funds are used significantly impacts donors' perceived efficacy and subsequent willingness to continue giving [27,28]. Donors' engagement and loyalty increase when they perceive their contributions to have real, visible impact. In China's "internet + public welfare" context, transparency of donation project information (especially traceability of fund usage), bureaucratic tendencies of fundraising entities, and interactive experiences during the donation process are critical variables influencing public willingness to donate [29,30]. Additionally, individuals' legal awareness, trust in fundraising entities, and ethical education profoundly affect their participation in philanthropy [31]. Comparative analysis of charitable fundraising legislation between China and the United States reveals that safeguarding citizens' freedom to solicit donations constitutes the essence of charitable fundraising protection, with the establishment of a well-designed post-event oversight system being the key to addressing challenges [32].

## 2.4 Research gap and analytical framework

Overall, existing research has conducted valuable explorations into the factors influencing fundraising effectiveness of charitable organizations across three dimensions: institutional environment, organizational capacity, and donor trust. These studies primarily focus on single, non-combined factors and employ qualitative or regression analysis methods, with few investigations examining the combined, complementary factors affecting fundraising effectiveness in charitable foundations. However, most events arise not from a single factor but from the combined effects of multiple factors [33]. To bridge this theoretical and methodological gap, a more scientific response is urgently needed to regard the factors influencing fundraising effectiveness of charitable foundations and their combined effects in the social field. Transcending

traditional linear thinking, this study employs the fuzzy set qualitative comparative analysis (fsQCA) method across multiple cases to explore the combined factors influencing fundraising effectiveness. It further incorporates the TOE (Technology-Organization-Environment) theoretical framework [34] to construct an analytical model. This framework posits that fundraising effectiveness is jointly shaped by three antecedent conditions: Technology, Organization, and Environment. It represents not a simplistic restatement of existing tripartite classifications but a more integrated framework that reorganizes and elevates core issues across these three dimensions. This provides a more comprehensive and practice-oriented theoretical explanation to advance the development of charitable foundations.

Based on literature review and analysis, this study categorizes factors influencing charitable foundation fundraising effectiveness under the TOE framework:

Environmental Dimension. This dimension integrates core issues of political connections and government resources in the "institutional environment" literature. It focuses on macro-level factors external to the organization that influence its operations and performance. In the Chinese context, the government serves not only as a regulator but also as the holder of critical resources and the ultimate grantor of legitimacy [26]. Establishing strong relationships with the government is a vital pathway to securing legitimacy and critical resources. Directly obtained government subsidies represent the direct economic manifestation of this legitimacy [35]. Therefore, this study selects Political Connections (PC) and government resource acquisition (GRA) as the core variables measuring the environmental dimension.

Organizational Dimension. This dimension incorporates discussions on professional management and program activity from the "organizational capacity" literature. This dimension focuses on the characteristics and abilities that can be controlled or constructed within the organization. In the highly competitive charitable market, foundations must rely on intrinsic capabilities to earn trust. Drawing on resource dependence theory, foundations must develop internal core competencies to manage resource channels. Professional Management Level (measured by full-time staff count) reflects an organization's human resource foundation and operational effectiveness, which determines its ability to translate resources into social impact [36]. Program Activity Intensity (measured by the number of projects) is a strategic asset for organizations to carry out business activities, send operational signals to the public, and establish stable connections with donors [24], which helps to manage external resource dependence. Therefore, this study selects Professional Management Level (PML) and Program Activity Intensity (PAI) as the core variables to measure the organizational dimension.

Technological Dimension. This dimension creatively transforms the micro mechanism of transparency, trust and technology adoption in the "donor trust" literature into an organization's strategic capacity to proactively employ information and communication technologies for information disclosure and trust-building. The term "technology" herein is construed in a broad sense, encompassing not only instrumental technologies such as digital payment, data analysis, and blockchain traceability, but also information communication approaches, digital operation platforms, and standardized disclosure procedures adopted by organizations to adapt to the digital philanthropy context and enhance fundraising efficiency. In the digital era, information disclosure and communication technologies constitute the core infrastructure for establishing the credibility of charitable foundations, which directly affects the formation and maintenance of donor trust. Existing research has verified that donor trust under the framework of "Internet + Public Welfare" is highly dependent on the transparency of charitable information and the traceability of fund utilization. Given data availability and compatibility with the research sample, this study adopts Digital Transparency Infrastructure (DTI) as the core measurement indicator of the technological dimension, defining it as the behavior of foundations in conducting proactive and standardized information disclosure through multiple channels including official websites, WeChat official accounts, and Sina Weibo. In essence, DTI captures the capability of foundations to adopt and apply modern information and communication technologies, proactively establish information transparency, respond to public concerns, and transmit credible signals to donors. Information disclosure behaviors under this dimension are no longer merely external supervision constraints, but rather "trust-building technologies" that foundations must master and actively employ in the digital philanthropy ecosystem, representing the core subset of the broad technological dimension that is most directly associated with fundraising efficiency

and quantifiable with empirical data. This study takes the information communication and disclosure technologies underpinning Digital Transparency Infrastructure as the core manifestation of the technological dimension, leaving room for future research to expand the measurement scope of the technological dimension.

In summary, the theoretical framework constructed in this study is illustrated in Fig 1:

## 3 Research design

### 3.1 Method selection

The Qualitative Comparative Analysis (QCA) method combines the dual advantages of qualitative and quantitative analysis. This method, grounded in set theory principles, employs conditional configuration analysis to investigate the factors influencing various events in real-world contexts. At its core, it explores which combinations of factors lead to the occurrence of a specific event or phenomenon. Its central argument posits that outcomes typically result from multiple antecedent conditions interacting in diverse ways—meaning multiple paths may converge toward the same result [37]. The fsQCA (fuzzy set) analysis method selected in the study is a category within QCA qualitative comparative analysis. The following factors are mainly considered when using this method: First, fuzzy set QCA (fsQCA) integrates both the category and degree of set membership. This enables more precise analysis and stricter consistency assessment within set theory—specifically, whether condition variables are both sufficient and necessary for outcome variables. This broadens the scope and depth of evaluation without exacerbating the problem of limited diversity [38]. Second, QCA analysis imposes its own constraints on sample size selection, with 10–60 cases being optimal based on organizational structural integrity and governance standardization. While small samples imply a significant portion of explanatory variables remain unobservable, as Lakin notes, limited variation is a common phenomenon in social science research, occurring even in large datasets [39]. Furthermore, fuzzy set qualitative comparative analysis (fsQCA) stands as one of the more efficient QCA methods for handling categories, set membership, and degree differentiation simultaneously [40].

### 3.2 Case and variable description

**3.2.1 Case selection and data sources.** Considering data reliability and accessibility, this study employed judgmental sampling method, taking the charitable foundations recorded by domestic civil affairs departments at all levels as the data acquisition objects. Data primarily originated from annual reports of charitable foundations published on the official

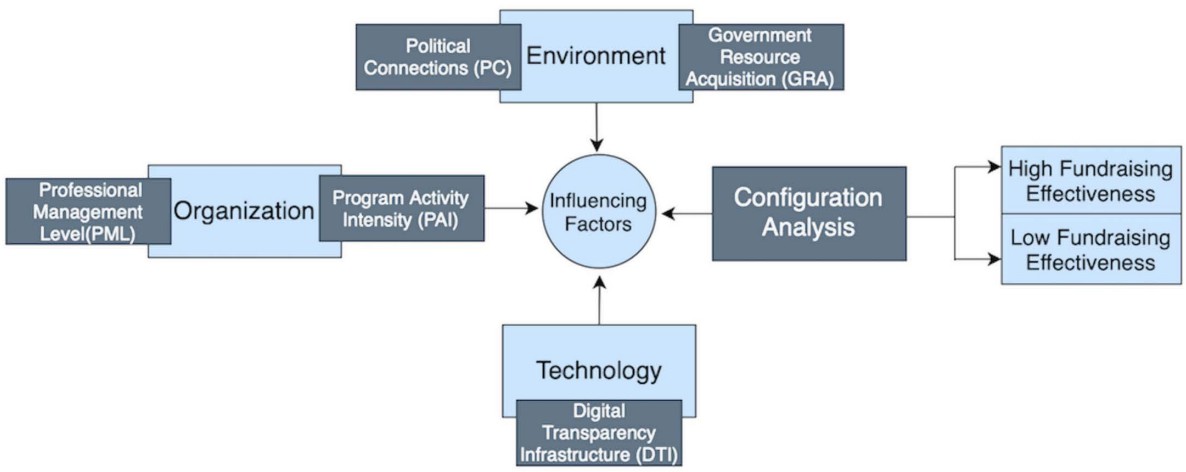

**Fig 1. Research Framework.**



website of China Social Organizations. A total of 80 charitable foundations providing nationwide services were selected. After excluding samples with missing data, 54 charitable foundations were retained as the final research sample, meeting the sample size criteria for QCA research.

**3.2.2 Outcome variable.** The outcome variable is Fundraising Effectiveness (FE), defined as the total annual public donation revenue of charitable foundations (excluding program service income, government purchase service income and other non-fundraising income). The data is extracted from the "donation income" line item in the annual financial reports of foundations, which strictly reflects the actual fundraising capacity of organizations (consistent with the universal definition of fundraising effectiveness in nonprofit research).

**3.2.3 Condition variables.** To avoid an unfavorable situation where the number of condition combinations far exceeds the sample size, thereby failing to reflect the actual situation, the selection of samples and variables strictly adheres to the selection principles of the QCA analysis method. When the number of case samples ranges from 10 to 60, it is considered a medium sample size, and the optimal number of condition variables should be maintained between 4 and 10. Based on previous studies and the TOE (Technology-Organization-Environment) framework established in the literature review, and considering data availability and completeness, this study defines the outcome variable and condition variables as follows. The outcome variable is Fundraising Effectiveness (FE). The condition variables are categorized into three dimensions: Environmental dimension: Political Connection (PC), Government Resource Acquisition (GRA); Organizational dimension: Professional Management Level (PML), Program Activity Intensity (PAI); Technological dimension: Digital Transparency Infrastructure (DTI). The specific measurement methods for each variable are detailed in Table 1:

Table 2 presents the results of descriptive statistics for all variables in the sample. The average fundraising income of 54 national charitable foundations was 37,949.04, with a standard deviation of 97,762.02. The range between the minimum value of 0 and the maximum value of 653,647.20 was substantial. The standard deviations for political connections and media supervision were relatively small, but the standard deviations for the conditional variables were generally large overall, especially for the government resource acquisition, which had a standard deviation of 13,989.47. The descriptive statistical analysis of variables is shown in Table 2:

### 3.3 Data calibration

Considering practical situations, the study primarily employs fuzzy calibration. At the same time, to ensure relatively objective calibration, the direct calibration method was adopted to perform quartile calibration on the antecedent conditions,

**Table 1. Variable Descriptions and Measurements.**

| Variable Abbreviation | Variable Name | Measurement Dimension | Theoretical Literature |
|---|---|---|---|
| FE | Fundraising Effectiveness | Total annual public donation revenue of charitable foundations | Zhang Y et al. (2020) [35] |
| PC | Political Connection | The number of current or retired government officials serving on the foundation's board of directors | Hillman (2003) [41] |
| GRA | Government Resource Acquisition | Refers to government funding subsidies, based on the annual government subsidy income amount reported in the foundation's annual report | Zhang Y et al. (2020) [35] |
| PML | Professional Management Level | Number of full-time staff members at the foundation | Zhang Y et al. (2020) [35] |
| PAI | Program Activity Intensity | Number of Public Welfare and Charitable Projects Undertaken by the Foundation This Year | Chen TX and Yao M (2012) [42] |
| DTI | Digital Transparency Infrastructure | Official website, WeChat public account, official Weibo, and publication of organizational information are the primary reference indicators. Each of these four points is worth 1 point and is added sequentially. | Zhang F (2021) [43] |

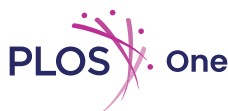

**Table 2. Descriptive Statistics Analysis of Variables.**

|  | FE | PC | GRA | PML | PAI | DTI |
|---|---|---|---|---|---|---|
| Average | 37,949.04 | 3.09 | 2785.25 | 24.09 | 29.61 | 2.87 |
| Standard Deviation | 97,762.02 | 3.12 | 13,989.47 | 30.34 | 30.78 | 0.73 |
| Minimum | 0 | 0 | 0 | 3 | 1 | 2 |
| Maximum | 653,647.20 | 15 | 100000.00 | 195 | 147 | 4 |

following the approach of Garcia Castro and Francoeur (2016) [44]. The study used fsQCA3.0 software to select the upper quartile (75%), median (50%), and lower quartile (25%) as the three anchor points for Fully in, Intersection Point, and Fully out [45]. It is worth noting that during the calibration of the conditional variable GRA, it was found that due to the influence of the distribution characteristics of the sample data, the intersection values set at the 50% percentile were set to 0, resulting in numerical overlap between the intersection points and Fully out points, making it difficult to form effective set discrimination. If the quantile method continues to be used, it may weaken the identification of the result variables in fuzzy set partitioning, which is not conducive to the stability of subsequent configuration analysis. Therefore, while maintaining the overall idea of quantile calibration, this article uses the mean as an auxiliary intersection point for calibration to ensure that the result set has a clear boundary between Fully in and Fully out. The calibration results are presented in Table 3:

Notably, the raw scores of the Digital Transparency Infrastructure (DTI) variable range from 2 to 4, with a mean of 2.87 and a standard deviation of 0.73. All sample foundations exhibit a certain level of digital transparency infrastructure, with no cases of complete absence. Fig 2 presents the distribution of the raw DTI scores, and Fig 3 displays the distribution of the calibrated fuzzy-set membership scores of DTI. As shown in Fig 3, the calibrated DTI membership scores exhibit a bimodal polarization: 18 cases have a membership score ≤ 0.2, and 36 cases have a membership score ≥ 0.8, with no observations in the intermediate range. This distribution pattern implies that DTI approximates a quasi-necessary condition in the sample, which may lead to a high consistency score in univariate necessity analysis. However, this is more likely to reflect a limitation of the sample structure rather than indicating that DTI independently constitutes a necessary condition for high fundraising income. Accordingly, this study focuses more on the joint effects of DTI with other factors within configurational conditions, rather than its independent explanatory role in terms of necessity. Histograms of the raw DTI scores and calibrated fuzzy-set membership scores of DTI are presented in Figs 2 and 3, respectively.

## 4 Data analysis

### 4.1 Univariate necessity and sufficiency analysis

The study first performed necessity and sufficiency analyses on each conditional variable before conducting conditional configuration analysis. Results were measured using two metrics: Consistency and Coverage. Consistency measures the degree of explanation, while Coverage indicates the number of cases that can be explained. The consistency and coverage between the conditional variable and the outcome variable can be calculated using the following two formulas:

**Table 3. Calibration of Result Variables and Condition Variables.**

| Variable Attribute | Variable Name | Fully in | Intersection Point | Fully out |
|---|---|---|---|---|
| Outcome Variable | Foundation Fundraising Effectiveness | 41,873.2 | 5,353.3 | 1121.4 |
| Condition Variable | Political Connections | 4 | 2 | 1 |
|  | Government Resource Acquisition | 150 | 2785.25 | 0 |
|  | Professional Management Level | 26.5 | 16 | 8 |
|  | Program Activity Intensity | 36 | 22 | 8 |
|  | Digital Transparency Infrastructure | 3 | 2.8703 | 2 |

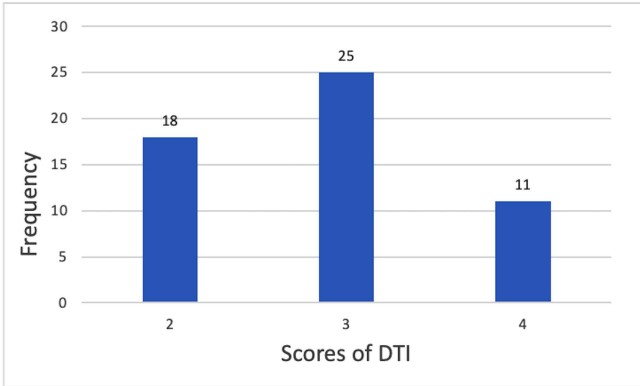

**Fig 2. Histogram of the raw DTI scores.**

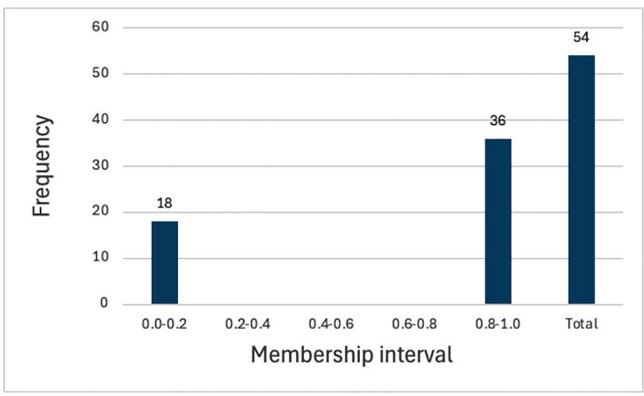

**Fig 3. Histogram of the calibrated fuzzy-set membership scores of DTI.**

Consistency (Xi ≤ Yi) = ∑ [min (Xi, Yi)]/∑Xi

Coverage (Xi ≤ Yi) = ∑ [min (Xi, Yi)]/∑Yi

The analysis was conducted using fsQCA 3.0 software, with results presented in Table 4:

Consistency measures the sufficient and necessary relationship between conditional variable X and outcome variable Y in geometric membership, while coverage measures the extent to which these given conditions or combinations of conditions explain the occurrence of the outcome. Ragin (2008) posits that consistency values range from 0 to 1, with a consistency score above 0.9 and coverage is greater than 0.5 indicating that the conditional variable is a necessary condition for the outcome [46]. Concurrently, if a single variable constitutes a sufficient condition for the outcome variable, its fuzzy set score must be less than or equal to that of the outcome variable, with a consistency index exceeding 0.8.

As shown in the results of single-factor necessity tests in Table 4, only strong Digital Transparency Infrastructure (DTI) presents a consistency score above 0.9 and coverage above 0.5, suggesting that strong DTI may be a necessary condition for foundations to achieve high fundraising income. In contrast, high professional management capability (0.882), high program activity intensity (0.877), and strong government resource acquisition capacity (0.832) all exceed the threshold



**Table 4. Necessity condition detection.**

| Condition Variable | Foundation High Donation Income | | Foundation Low Donation Income | |
| --- | --- | --- | --- | --- |
| | Consistency | Coverage | Consistency | Coverage |
| Strong Political Connection | 0.725 | 0.647 | 0.845 | 0.513 |
| Weak Political Connection | 0.455 | 0.811 | 0.419 | 0.509 |
| Strong Government Resource Acquisition | 0.832 | 0.755 | 0.799 | 0.494 |
| Weak Government Resource Acquisition | 0.444 | 0.765 | 0.605 | 0.710 |
| Highly Professional Management Level | 0.882 | 0.791 | 0.650 | 0.397 |
| Low Professional Management Level | 0.327 | 0.579 | 0.657 | 0.791 |
| High Program Activity Intensity | 0.877 | 0.765 | 0.636 | 0.378 |
| Low Program Activity Intensity | 0.287 | 0.537 | 0.605 | 0.770 |
| Strong Digital Transparency Infrastructure | 0.908 | 0.667 | 0.923 | 0.461 |
| Weak Digital Transparency Infrastructure | 0.266 | 0.835 | 0.333 | 0.710 |

of 0.8, indicating that these four variables represent sufficient but not necessary conditions for variations in fundraising performance among charitable foundations. However, the consistency values of all other variables are below 0.75, implying that these variables cannot independently explain changes in fundraising income. This necessitates configurational analysis to identify combinations of multiple factors driving variations in fundraising income.

In fsQCA, consistency ≥ 0.9 and coverage > 0.5 are commonly used as criteria for identifying a necessary condition, while consistency ≥ 0.8 serves as a reference threshold for assessing the sufficiency of a single condition. It should be emphasized that valid judgments of sufficiency rely on truth table and configurational analysis rather than single-condition consistency alone, which constitutes the core distinction between necessity and sufficiency analyses. As shown in Table 4, strong DTI yields a consistency score of 0.908 for high fundraising performance, which meets the numerical benchmark for a necessary condition. However, its consistency for low fundraising performance reaches 0.923, with no significant difference between the two values. This indicates that DTI lacks discriminatory power between high and low fundraising performance, requiring further verification of its necessity via visual analysis.

To further examine the necessity of individual conditions, this study presents XY plots of necessity analyses for DTI with respect to high fundraising performance (Y) and low fundraising performance (~Y) (Figs 4 and 5). A core criterion for identifying a necessary condition in fsQCA is that the fuzzy-set membership of the condition is always less than or equal to that of the outcome, which appears as a concentration of observations in the upper-left region above the 45° diagonal in XY plots.

As shown in Fig 4, observations for DTI relative to high fundraising performance do not cluster noticeably in the upper-left region; instead, many points lie near the vertical Y-axis, thus failing to satisfy the distributional requirement for a necessary condition. Similarly, Fig 5 reveals no upper-left clustering for DTI relative to low fundraising performance. Combined with the results in Table 4, DTI yields a consistency of 0.908 for high fundraising performance and 0.923 for low fundraising performance, with the latter even slightly higher. This demonstrates that DTI is prevalent across the sample and lacks discriminatory power between high and low fundraising outcomes and therefore is not a necessary condition for achieving high fundraising performance.

Furthermore, no other single condition reaches the 0.9 consistency threshold for necessity, and single-condition consistency cannot be used to infer sufficiency. Accordingly, this study does not identify any single condition as either necessary or sufficient for high fundraising performance. Instead, all conditions are treated as potential explanatory variables and incorporated into subsequent configurational analysis to explore their combined effects.

The XY plots for the necessity analysis of Digital Transparency Infrastructure (DTI) with respect to high fundraising performance (Y) and low fundraising performance (~Y) are presented in Figs 4 and 5, respectively.

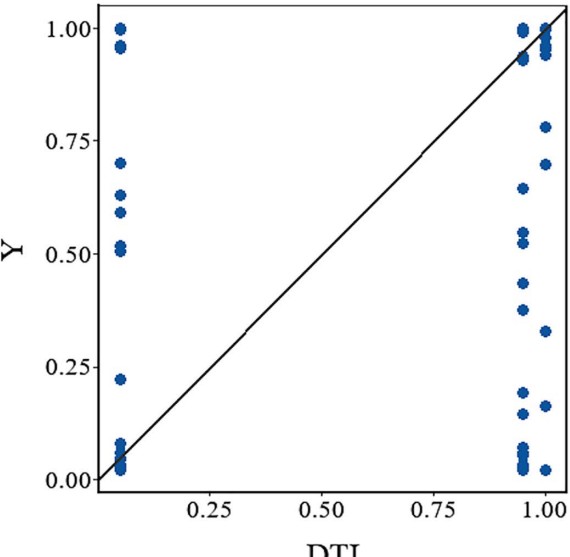

**Fig 4. XY plot for necessity analysis of Digital Transparency Infrastructure (DTI) with respect to high fundraising performance (Y).**

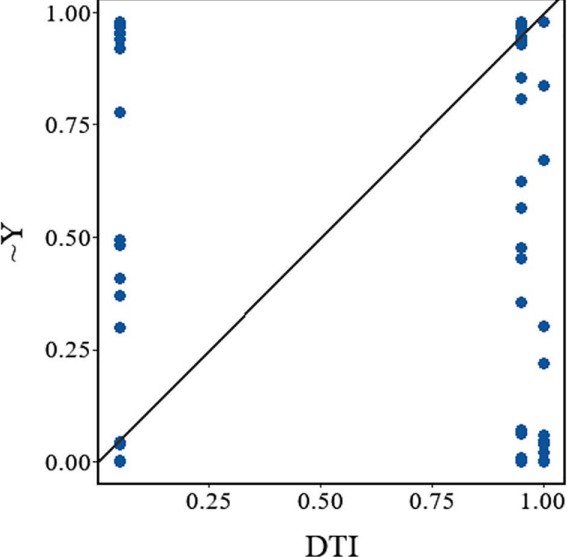

**Fig 5. XY plot for necessity analysis of Digital Transparency Infrastructure (DTI) with respect to low fundraising performance (~Y).**

### 4.2 Conditional configuration analysis

Prior to conducting the configurational analysis, a truth table was constructed using fsQCA 3.0 software (Table 5). Given the relatively large number of condition variables in this study and the varying frequencies of different condition combinations across the sample, the indiscriminate inclusion of all combinations may introduce a large number of logical

remainders lacking empirical support, thereby undermining the robustness and explanatory power of the configurational results. Therefore, it is necessary to reasonably set the thresholds for case frequency and consistency. Considering the total sample size of 54 in this study and following conventional practices in existing fsQCA studies for medium-sized samples, the case frequency threshold was set to 1 to maximize the retention of sample information and avoid the loss of valid configurations. For the consistency threshold, the academic community generally agrees that consistency should not be lower than 0.75, and higher values indicate stronger sufficiency of the condition configurations for the outcome variable. Integrating the theoretical expectations and sample characteristics of this study, and referring to relevant literature, the raw consistency threshold was set to 0.8, and the PRI consistency threshold was set to 0.7 [47]. This setting helps to eliminate simultaneous subset relations between configurations, avoid interpretive biases caused by asymmetric causal relationships, and improve the rigor of the analytical results. The truth table is shown in Table 5:

According to the truth table for QCA data statistics, Fiss (2011) uses solid circles to represent the presence of conditions and hollow circles to represent the absence of conditions in order to visually report the results [48]. Circle size distinguishes core conditions from peripheral conditions, while blank spaces indicate optional conditions. Core conditions are defined as those that appear in both the parsimonious solution and the intermediate solution and are critical for the outcome. Peripheral conditions refer to those that appear only in the intermediate solution in the QCA analysis and are less important for the outcome compared with core conditions. The complex solution presents the most conservative configurational relationships. The intermediate solution, which incorporates limited counterfactuals based on theoretical expectations, serves as the primary interpretive basis in this study. Referring to the QCA result presentation methods proposed by Ragin (2008) [46] and Fiss (2011) [48], the findings are visualized in Table 6:

Table 6 presents two distinct explanatory paths, each constituting a subset influencing the fundraising effectiveness of foundation organizations. These paths explain the reasons behind high fundraising effectiveness in foundation organizations. Their combined total coverage of 0.803 indicates that approximately 80% of cases involving foundation organizations with high fundraising effectiveness can be explained. The combined paths shown in this table are illustrated in Table 7:

As shown in Table 7, there are two pathways for foundations to achieve better fundraising effectiveness:

Path 1: Strong Political Connections * High Program Activity Intensity * Strong Digital Transparency Infrastructure.

Path 2: High Professional Management Level * High Program Activity Intensity * Strong Digital Transparency Infrastructure.

Optimal fundraising effectiveness = High Program Activity Intensity * Strong Digital Transparency Infrastructure * (Strong Political Connections + High Professional Management Level).

**Table 5. Truth table.**

| PC | GRA | PML | PAI | DTI | FE | Number |
|---|---|---|---|---|---|---|
| 1 | 1 | 0 | 0 | 1 | 0 | 4 |
| 1 | 0 | 0 | 0 | 1 | 0 | 1 |
| 1 | 0 | 1 | 0 | 1 | 0 | 1 |
| 1 | 1 | 1 | 0 | 1 | 0 | 2 |
| 1 | 0 | 0 | 1 | 1 | 1 | 1 |
| 1 | 1 | 0 | 1 | 1 | 1 | 2 |
| 1 | 0 | 1 | 1 | 1 | 1 | 3 |
| 1 | 1 | 1 | 1 | 1 | 1 | 14 |
| 0 | 1 | 1 | 1 | 1 | 1 | 6 |
| 0 | 0 | 1 | 1 | 1 | 1 | 1 |

**Table 6. Configuration Analysis of High Fundraising Effectiveness in Foundations.**

| Condition | Configuration Solution | |
| --- | --- | --- |
| | **Path Combination 1** | **Path Combination 2** |
| Political Connections | • | — |
| Government Resource Acquisition | — | — |
| High Professional Management Level | — | • |
| Program Activity Intensity | ● | ● |
| Digital Transparency Infrastructure | • | • |
| Consistency | 0.826 | 0.887 |
| Original Coverage | 0.609 | 0.769 |
| Unique Coverage | 0.034 | 0.194 |
| Total consistency | 0.843 | |
| Total Coverage | 0.803 | |

Symbolic representation is as follows: Note: "●" and "•" indicate the presence of a condition, meaning the condition variable has a high value. "●" denotes a core condition, "•" denotes a peripheral condition, and "—" indicates that the condition variable has no effect on the outcome.

**Table 7. Optimal Combined Paths.**

| Causal Combination | Original Coverage | Net Coverage | Consis-tency |
| --- | --- | --- | --- |
| Strong Political Connections*High Program Activity Intensity*Strong Digital Transparency Infrastructure | 0.609 | 0.034 | 0.826 |
| High Professional Management Level*High Program Activity Intensity*Strong Digital Transparency Infrastructure | 0.769 | 0.194 | 0.887 |
| Solution Coverage | 0.803 | | |
| Solution consistency | 0.843 | | |

## 4.3 Configuration path analysis

Five factors influence fundraising performance in the fundraising process of charitable foundations. Among them, digital transparency infrastructure and program activity intensity are the core conditions affecting fundraising performance, while professional management level and political connection exert weaker effects than the former two conditions. These factors can only exert their influence when combined with strong digital transparency infrastructure and high program activity intensity. In addition, the overall consistency is 0.843 and the overall coverage is 0.803, both of which exceed the conventional thresholds for fsQCA analysis. This indicates that the empirical results are valid and possess strong explanatory power for the sufficiency of configurations leading to high fundraising performance. The conditional configurations are illustrated in Fig 6:

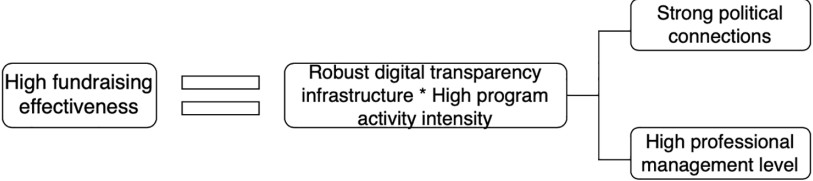

**Fig 6. Path Analysis of Condition Configurations.**

Condition Configuration 1 indicates that strong political connections combined with robust digital transparency infrastructure and high program activity intensity enable charitable foundations to achieve superior fundraising effectiveness. This suggests that when a foundation possesses strong political connections, coupled with high program activity intensity and well-established digital transparency infrastructure, its fundraising activities can meet desired goals even if the foundation lacks sufficient capacity to draw on government resources. Furthermore, the original coverage rate of Configuration 1 reaches approximately 0.61, indicating that about 61% of successful fundraising cases among foundations can be explained by this causal pathway. The unique coverage was 0.034, meaning only about 3% of successful fundraising cases could be explained solely by this causal pathway. Previous research has also found that political connections serve as an intermediary for charitable foundations to obtain administrative legitimacy, playing a significant role in the process of securing government permits [49]. With the rapid development of the media era, the diversity of information disclosure channels directly influences public willingness to donate to charitable organizations. That is, the accessibility of media supervision in the era of all media has become a necessary condition for foundations to achieve high fundraising performance [50].

Condition Configuration 2 indicates that the synergistic combination of high program activity intensity, strong digital transparency infrastructure, and highly professional management constitutes a key pathway for charitable foundations to achieve high fundraising effectiveness. This suggests that when foundations possess high program activity intensity, robust digital transparency infrastructure, and advanced professional management, their fundraising effectiveness can reach an ideal state even with low political connections and weak government resource acquisition. Furthermore, the original coverage and net coverage of configuration two are the highest in the configuration, at 0.769 and 0.194, respectively. This indicates that approximately 77% of successful fundraising effectiveness cases can be explained by this causal pathway, while about 19% of successful fundraising cases can only be successfully explained by this pathway. In China, "number of full-time staff" has been incorporated into strategic development indicators for social organizations (nonprofit organizations). This signifies that the professional management level of charitable foundations will serve as a crucial benchmark for assessing their developmental status for the foreseeable future. Meanwhile, the powerful effect of high program activity intensity in promoting sustainable and healthy organizational development has been validated by countless real cases [51]. Charitable foundations seeking to maintain a stable and leading development trajectory within the industry must therefore strengthen the intensity of project activities

Finally, further analysis was conducted on the absence of high fundraising effectiveness and the causal asymmetry of its influencing mechanisms in charitable foundations, exploring the "causal asymmetry" of the influencing mechanism. The results are shown in Table 8:

In the condition configuration for low fundraising effectiveness, overall consistency and overall coverage reached 0.79 and 0.65 respectively, both exceeding the threshold. Condition Configuration 3 comprises strong political connections, strong digital transparency infrastructure, and low program activity intensity. Condition configuration 4 comprises strong political connections, low professional management levels, and strong digital transparency infrastructure. The original coverage of these two conditional configurations is comparable, both above 0.5, indicating that these two causal paths can each explain about 50% of the fundraising effectiveness of charitable foundations. The results reveal that low fundraising returns in charitable foundations are also influenced by the combined effects of political and external oversight dimensions. Different combinations of political connections with digital transparency infrastructure, professional management levels, and program activity intensity not only do not lead to low fundraising returns but can also explain high fundraising effectiveness. Therefore, analyzing the combined effects of various influencing factors is essential when considering fundraising returns in charitable foundations. Simultaneously, the findings confirm that under the logical premise of "causal asymmetry", the conditions leading to high fundraising returns for charitable foundations do not correspond one-to-one with those causing low fundraising returns, indicating asymmetry. The formation of this asymmetric feature is closely associated with the systematic resource allocation bias caused by the Matthew effect in the charitable sector. Under the

**Table 8. Low Effectiveness Configuration Analysis of Foundation Fundraising.**

| Condition | Configuration Solution | |
|---|---|---|
| | **Pathway Combination 3** | **Path Combination 4** |
| Political Connections | • | • |
| Government Resource Acquisition | — | — |
| Professional Management Level | — | ⊗ |
| Program Activity Intensity | ⊗ | — |
| Digital Transparency Infrastructure | • | • |
| Consistency | 0.800 | 0.812 |
| Original Coverage | 0.514 | 0.528 |
| Unique Coverage | 0.123 | 0.136 |
| Total Consistency | 0.790 | |
| Total Coverage | 0.650 | |

Note: "⊗" indicates the condition does not appear; both denote core conditions. "•" indicates peripheral conditions. "—" indicates the condition variable has no effect on the outcome. Core conditions refer to condition variables appearing simultaneously in both the "minimal solution" and "intermediate solution" of QCA analysis, carrying greater significance for the outcome variable.

influence of the Matthew effect, foundations with low fundraising performance are highly prone to falling into a vicious cycle: although such foundations may also possess the basic conditions of strong political connections and robust digital transparency infrastructure, their critical shortcomings—insufficient program activity intensity or weak professional management level—prevent them from translating existing conditions into actual fundraising capabilities. The persistent downturn in fundraising performance further deprives foundations of critical resources such as capital and talent to strengthen program activities and improve professional management. This ultimately results in a configurational logic distinct from that of high-fundraising-performance foundations. It also indicates that merely adjusting a single condition in reverse cannot effectively resolve the development dilemma of low fundraising performance among charitable foundations.

### 4.4 Robustness test

QCA robustness test encompasses multiple approaches, with the common method involving legitimate adjustments to parameter settings. If parameter adjustments do not result in substantial changes to the number, composition, consistency and coverage of configurations, the analysis results can be considered reliable [52]. During robustness testing, the consistency threshold was adjusted from 0.8 to 0.85. The calculated number of configurations and their components remained consistent with the original results. Overall consistency remained approximately 0.87, still above the critical threshold. In summary, the analysis results obtained from robustness testing did not undergo substantial changes compared to previous analysis results, indicating the stability of the previous research conclusions in this article.

## 5 Conclusions and implications

### 5.1 Main conclusions

Resource scarcity amid expansion aspirations has become a critical bottleneck for charitable organizations to overcome. However, acquiring development resources and enhancing fundraising effectiveness requires these organizations to actively explore solutions across political, organizational, and social dimensions. Research findings indicate two successful pathways for charitable foundations to achieve high fundraising effectiveness: first, the combined effect of comprehensive digital transparency infrastructure, high project activity intensity, and strong political connections; second, the



combined effect of comprehensive digital transparency infrastructure, high project activity intensity, and high professional management levels. Compared to other factors, comprehensive digital transparency infrastructure and high project activity intensity are the most critical factors for charitable organizations to successfully achieve high fundraising returns. To trace the cause, the fundraising effectiveness of charitable organizations hinges on public trust, with digital transparency infrastructure and project activity intensity serving as the cornerstones for building this trust foundation. Comprehensive digital transparency infrastructure, acting as an external constraint mechanism, compels organizations to standardize information disclosure processes and enhance transparency in fund usage, directly boosting public trust. Simultaneously, the social exposure generated by digital transparency infrastructure expands the organization's visibility, enabling more donors to perceive its philanthropic value. This proactively cultivated visibility serves as the technical infrastructure for building organizational credibility in the digital age. A high project activity intensity not only helps the public quickly establish a clear understanding of the organization's mission and operational scope, reducing decision-making costs when selecting donation organization, but also helps organizations build a positive reputation and cultivate a stable, loyal donor group. The two together form the underlying foundation for improving the fundraising effectiveness of charitable foundations. Political connections and professional management level serve as auxiliary paths to further amplify the ability to acquire resources. Without these two core elements, other factors struggle to convert effectively into tangible fundraising outcomes. Based on this analysis, the study proposes the following recommendations to guide charitable organizations in improving fundraising effectiveness:

First, optimize organizational political connections to enhance government resource acquisition capabilities. Charitable organizations' political affiliations are often viewed by civil society regulators as barometers of state-society relations. According to resource dependency theory, foundations with strong political connections experience positive effects on government resource acquisition, specifically by promoting the standardization and legalization of internal management, thereby expanding the cooperation network with the government. Foundations with current or former government officials on staff can leverage these networks to actively participate in government tenders, forging high-quality cooperative relationships that enhance their capacity to access government resources. This also improves the organization's absorption capacity for government resources, providing financial support for social resource mobilization activities.

Second, foundations should strengthen the intensity of program activities. As a vital strategy for legitimacy and reputation building in the nonprofit sector, program activities of charitable foundations have been widely validated in both practice and existing research to enhance organizational visibility, donor recognition, and ultimately fundraising effectiveness. Foundation managers should incorporate it into organizational development strategies. This fosters strong "loyalty" bonds between the foundation and its audience, shapes a distinctive organizational image, and builds a solid public foundation for future fundraising activities. It also creates a positive word-of-mouth chain, significantly reducing the promotional costs associated with fundraising efforts. Additionally, research indicates that when a foundation possesses both comprehensive digital transparency infrastructure and high project activity intensity, enhancing professional management standards and ensuring a steady influx of specialized talent can further elevate the effectiveness of its fundraising activities.

Third, establish a transparent and traceable supervision system. In recent years, the "National Social Organization Official Website Protection Plan" jointly organized by civil affairs departments and authoritative internet companies has been systematically implemented. This initiative aims to provide nonprofit organizations with a fertile ground for growth, ensuring "digital empowerment" is fully integrated into all aspects of social organizations. The "Internet Plus Fundraising" model, centered on online platforms, is increasingly prominent. Internet users have developed their own information screening systems and demand high transparency and authority in information. To gain the favor of such "network specialists", the organizational structure of the foundation must establish a comprehensive information disclosure system that combines integrity and timeliness, providing real-time updates on daily activities and internal governance dynamics. These updates may include adjustments to organizational management policies, progress on activity projects, or publicity regarding participation in government initiatives. Such measures help netizens comprehensively and promptly monitor the development

status of target organizations, thereby expanding their audience reach. As key innovators in the third distribution mechanism, charitable foundations shoulder the vital mission of advancing philanthropy. However, each organization starts from different points, develops at varying speeds, and accesses resources differently. This necessitates that organizations select development pathways suited to their capabilities and needs while minimizing costs. Organizational development paths emphasize "adaptability"; identifying a path aligned with the foundation's developmental requirements is crucial for its sustainable and healthy development.

### 5.2 Limitations of the study

This study has several limitations. First, in terms of sample size, this study adopts a medium sample design consisting of 54 cases, which is consistent with the recommended sample range for fsQCA research. However, the relatively limited sample restricts the observation of some potential explanatory variables, including micro-level organizational operational practices, regional philanthropic culture, and the depth of digital platform application. Therefore, this study may not fully capture the heterogeneity in fundraising efficiency among different foundations, nor does it comprehensively examine the interplay between contextual factors and configurational paths. Second, regarding variable coverage, the present study operationalizes the technological dimension using information disclosure practices. Although this is consistent with data availability, it narrows the concept of "technology" to a subset of digital tools. Future research should incorporate a broader range of digital technologies, such as artificial intelligence-driven donor matching, blockchain traceability, and data analytics-enabled fundraising optimization, to evaluate the impact of technology more comprehensively. Third, this study focuses on national-level charitable foundations. Given the pronounced Matthew effect in China's philanthropic sector (i.e., resources are concentrated toward mature foundations), the research findings may not be fully generalizable to local foundations. Future studies should expand the sample to include municipal- and county-level foundations to explore cross-level configurational differences.

### 5.3 Future research directions

First, expand the technological dimension. This study only takes digital transparency infrastructure as the core of the technological dimension, focusing on the impact of foundations' information disclosure capacity on fundraising performance. Against the backdrop of digital philanthropy development, technological applications such as artificial intelligence-driven donor matching, blockchain-enabled fund traceability, and big data analytics-empowered precise fundraising are profoundly reshaping the operational model of charitable fundraising and donor experience. Follow-up studies can incorporate such intelligent technology applications into the analytical framework, explore their influencing paths on donor interactive experience, and the internal mechanism through which interactive experience further acts on donation loyalty and perceived donation effectiveness. In addition, future research can excavate new configurational paths for technology-enabled improvement of fundraising performance of charitable foundations to adapt to the development needs of philanthropy in the digital era.

Second, adopt longitudinal and dynamic analyses. This study adopts a cross-sectional fsQCA design, which limits the exploration of temporal dynamics. Future research should conduct longitudinal tracking to analyze the evolution of configurational paths at different developmental stages, as well as the temporal changes in the relative importance of core conditions. Dynamic analysis can enhance the explanatory power of the formation mechanism of long-term fundraising efficiency.

Third, compare national and local foundations. To address the Matthew effect and sample representativeness issues, future research should include municipal- and county-level local foundations. A comparative analysis between national and local foundations can reveal differences in high-performance configurational paths, as well as the moderating effects of regional philanthropic culture, government support and local institutional environment.

Fourth, classify foundations by type and explore political connections. Future studies can divide foundations into government-associated foundations and privately initiated foundations. This classification helps to reveal the differentiated role of political connections: whether political connections shift from peripheral conditions to core conditions in government-associated foundations, and whether the intensity of professional management and program activity intensity is further enhanced in privately initiated foundations. Meanwhile, research can explore the interactive effects between political connections and market resources.

Fifth, investigate failure avoidance in low-performance paths. Based on the observed causal asymmetry, future research should focus on failure modes and avoidance mechanisms. Specifically, low-performance paths characterized by complete digital transparency infrastructure＋strong political connections＋low program activity intensity and complete digital transparency infrastructure＋strong political connections＋low professional management can be analyzed. Through comparative case studies, researchers can identify strategies for performance improvement and develop early warning models to help foundations avoid configurational traps.

## Author contributions

**Conceptualization:** Zhe Zhu, Shan Huang, Heran Zou.

**Data curation:** Zhe Zhu, Shan Huang, Wanqiang Xu.

**Formal analysis:** Shan Huang.

**Funding acquisition:** Zhe Zhu.

**Investigation:** Zhe Zhu, Shan Huang, Wanqiang Xu.

**Methodology:** Zhe Zhu, Shan Huang, Heran Zou.

**Project administration:** Wanqiang Xu, Heran Zou.

**Resources:** Zhe Zhu.

**Software:** Zhe Zhu, Shan Huang.

**Supervision:** Wanqiang Xu, Heran Zou.

**Validation:** Wanqiang Xu.

**Writing – original draft:** Zhe Zhu, Shan Huang.

**Writing – review & editing:** Heran Zou.

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
