## [Decision Letter · Decision Letter 0]

8 Feb 2026

PONE-D-25-66814Influencing Factors and Configuration Pathways of Fundraising Effectiveness in Charitable Foundations: An FSQCA Analysis of 54 Cases in ChinaPLOS One

Dear Dr. Zou,

Thank you for submitting your manuscript to PLOS ONE. After careful consideration, we feel that it has merit but does not fully meet PLOS ONE’s publication criteria as it currently stands. Therefore, we invite you to submit a revised version of the manuscript that addresses the points raised during the review process.

We look forward to receiving your revised manuscript.

Kind regards,

Abel C. H. Chen

Academic Editor

PLOS One

Journal Requirements:

2. We note you have included a table to which you do not refer in the text of your manuscript. Please ensure that you refer to Table 1 in your text; if accepted, production will need this reference to link the reader to the Table

3. Please ensure that you refer to Figure 1 in your text as, if accepted, production will need this reference to link the reader to the figure.

4. Please note that your Data Availability Statement is currently unable to open a direct link to access each database. If your manuscript is accepted for publication, you will be asked to provide these details on a very short timeline. We therefore suggest that you provide this information now, though we will not hold up the peer review process if you are unable.

Reviewers' comments:

Reviewer's Responses to Questions

**Comments to the Author**

1. Is the manuscript technically sound, and do the data support the conclusions?

Reviewer #1: Yes

Reviewer #2: Yes

2. Has the statistical analysis been performed appropriately and rigorously? 

Reviewer #1: Yes

Reviewer #2: Yes

3. Have the authors made all data underlying the findings in their manuscript fully available?

Reviewer #1: Yes

Reviewer #2: Yes

4. Is the manuscript presented in an intelligible fashion and written in standard English?

Reviewer #1: Yes

Reviewer #2: Yes

5. Review Comments to the Author

Reviewer #1: The manuscript applies fsQCA (54 cases) to identify configurations leading to high/low fundraising effectiveness (FE) in Chinese charitable foundations. It operationalizes antecedent conditions under a TOE (Technology–Organization–Environment) framing: Political Connections (PC), Government Resource Acquisition (GRA), Professional Management Level (PML), Brand Building Level (BBL), and Media Supervision (MS). The headline finding is that MS and BBL are “core” conditions for high FE, with two sufficient configurations: (1) BBLMSPC and (2) BBLMSPML.

The topic is relevant, and a configurational approach can be appropriate. However, the current paper has several publication-critical issues: (i) outcome and condition measurement validity (several constructs are mis-specified or conflated), (ii) calibration choices contain internal inconsistencies (notably GRA), (iii) truth-table design appears to collapse variation (MS is nearly constant), (iv) interpretation overstates necessity/sufficiency and “core” claims, and (v) sampling and reproducibility do not meet PLOS ONE standards. I recommend Major Revision.

Major Comments

1. Construct validity problems: several key variables do not measure what the manuscript claims

1.1 Fundraising Effectiveness (FE) is defined as “aggregate revenue generated through project activities and providing public goods/services” (Methods 3.2.2; Table 1). This is not clearly “fundraising effectiveness” as commonly understood. It mixes: fundraising income vs program/service revenues, scale vs efficiency, and possibly government/purchased services if booked as project income.

If the construct is “fundraising effectiveness,” the outcome should be operationalized as: total donations received; public fundraising revenue; online donation amount; or a ratio-based metric (e.g., fundraising revenue per staff, per project, or as share of total revenue). At minimum, you must clarify the accounting category used in annual reports (which line item), and justify why it maps to “fundraising effectiveness.”

1.2 “Brand Building Level (BBL)” is measured as the “number of public welfare and charitable projects undertaken this year.”

This is closer to program activity volume/scale than brand building. Using project count as “brand building” risks circularity: more projects can mechanically generate more “project revenue” if FE is tied to program activities. That can create an artificial configuration where “BBL” predicts “FE” because both reflect activity scale.

Suggested fix: either re-label BBL as “program activity intensity/portfolio breadth” and adjust theory, or construct a genuine brand proxy: brand mentions, media exposure metrics, third-party ratings, social media followers/engagement, donor retention, website traffic, or reputational awards.

1.3 “Media Supervision (MS)” is operationalized as a 2–4 score based on existence of website/WeChat/Weibo and information disclosure.

This is not media supervision; it is digital presence + disclosure practice, and it is largely an organizational capability rather than external supervision. The paper’s interpretation (MS as an external constraint) conflicts with its measurement (internal adoption).

Suggested fix: rename to “digital transparency infrastructure” (or “information disclosure capacity”), and if you truly want “media supervision,” measure negative/positive media coverage, scrutiny events, or exposure to investigative reporting.

2. Calibration is internally inconsistent; GRA thresholds appear invalid and will distort results

Table 3 sets GRA calibration as Fully in = 150, Intersection = 0, Fully out = 0. This is not a proper three-anchor calibration because the cross-over and full-out are identical (0). With many zeros in GRA (Table 2 min=0), this will force a large mass of cases to extreme membership and can create spurious necessity/sufficiency patterns.

Moreover, the text states anchors are set at 25/50/75 percentiles (Garcia-Castro & Francoeur 2016), but the GRA anchors do not look like 25/50/75 percentiles given mean 2785 and max 100000. This suggests either (i) misreporting, or (ii) using a different rule only for GRA.

Required revision:

1) Report the empirical 25th/50th/75th percentiles for each raw variable and show how anchors were chosen.

2) Consider log-transforming highly skewed monetary variables before calibration.

3) Use a nonzero cross-over for GRA (e.g., median) and distinct full-out (e.g., 10th/25th percentile), or adopt theoretically justified anchors.

3. Limited diversity and quasi-constant conditions: MS has minimum 2 and very low variance

Descriptives show MS ranges from 2 to 4, mean 2.87, SD 0.73, min 2. This implies most cases already satisfy “some MS,” and no case is truly “low MS” in raw terms. Yet the necessity test claims “Strong media supervision” is necessary for high FE (consistency 0.908).

If MS is near-constant, it can appear necessary simply because it rarely varies and is present for almost everyone. This is a classic QCA pitfall: necessity can be an artifact of low variance.

Required revision:

1) Provide the distribution of raw MS scores and calibrated memberships (histogram or frequency table).

2) Test whether the necessity claim holds under alternative MS calibration anchors.

3) Consider replacing MS with a more discriminating measure (e.g., actual disclosure quality indices; frequency of updates; third-party transparency ratings).

4. Truth table construction and labeling issues suggest errors in analysis pipeline

In Table 5 (Truth Table), the columns list PC GML PMl BPL MS but earlier you define GRA and PML; “GML/PMl” appear inconsistent typos.

Additionally, Table 5 shows FE=0 for several rows even when MS=1 and PC=1 etc., but the manuscript then states it uses the “complex solution” to exclude counterfactuals. The procedure needs to be described precisely: frequency threshold (how many cases per row retained), consistency threshold for sufficiency, PRI consistency (to avoid simultaneous subset relations).

Required revision:

1) Provide the full fsQCA settings: frequency cutoff, consistency cutoff, PRI cutoff, directional expectations (intermediate solution), and how contradictions were handled.

2) Include the intermediate and parsimonious solutions (not only complex), because “core vs peripheral” is defined relative to parsimonious/intermediate comparisons. The current presentation risks misclassifying “core” conditions.

5. Interpretation problems: “necessary” vs “sufficient,” and “core” claims are overstated

You assert that MS is a necessary condition for high FE because consistency >0.9 (0.908). However, standard practice is to also examine: whether necessity holds for the negated outcome (it appears MS is also high for low FE: 0.923), which undermines practical necessity, relevance-of-necessity metrics (RoN) and empirical coverage logic, and trivial necessity due to high prevalence.

In Table 4, “Strong media supervision” has consistency 0.908 for high FE but also 0.923 for low FE. This is a red flag: a condition that is “necessary” for both outcome and non-outcome is not discriminating; it may simply be ubiquitous.

Also, the manuscript claims PML, BBL, and GRA are “sufficient but not necessary” based on their necessity-table consistency exceeding 0.8. That is not a valid inference: a necessity-table consistency >0.8 does not establish sufficiency; sufficiency must be evaluated using sufficiency analysis/truth table solutions for X→Y, not by necessity metrics.

Required revision:

1) Separate necessity analysis from sufficiency analysis properly.

2) Add XY-plots for key necessity claims.

3) Avoid “core element” language unless the core/peripheral status is methodologically supported and robust to specifications.

Reviewer #2: IMHO The manuscript requires corrections in terminological consistency, structural redundancy, and the idiomatic clarity of its academic prose. While the research design is theoretically sound, the communication of these ideas often suffers from translation-style artifacts and repetitive phrasing.

RESEARCH MERITS TO BE IMPROVED

- Narrow Variable Definition: The "Technological Dimension" in the TOE framework is operationalized almost exclusively as "media supervision". Critics may argue that "technology" should encompass broader infrastructure, such as digital payment systems or data analytics, rather than just information disclosure behavior

- The study uses 54 cases, which the authors justify as a "medium sample size" optimal for QCA. However, they acknowledge that this size leaves a significant portion of explanatory variables unobservable, which could be more explicitly addressed as a limitation in the conclusion.

- Contextual Depth on the "Matthew Effect": The paper mentions a significant "Matthew effect" (where successful organizations attract even more resources) in Chinese foundations but does not fully integrate how this systemic bias might influence the "causal asymmetry" found in the fsQCA results

- IMHO it is also necessary to suggest possible development paths in the Conclusions section.

The development paths could possibily focus on expanding the "Technology" dimension, transitioning to longitudinal data analysis, and investigating the heterogeneity of foundation types beyond the national level. For example, based on "Matthew effect" and the current sample limitation the following research opportunities could be identified:

-- conceptual expansion if the Technological Dimension through adoption of AI-driven donor matching, blockchain for fund traceability, and big data analytics and exploring how interactive experiences during the donation process—rather than just information disclosure—impact loyalty and perceived efficacy,

-- transition to temporal and dynamic analysis of timeshifts of different pathways esp. highly effective pathways,

-- comparison of causal paths for success for National-wide and local foundations,

-- investigation of difference in roles played by political connections: governmental-linked foundations vs. purely private initiatives,

-- assymetry of failure investigation e.g. to help fundation managers to avoid it.

There is so much to do in the future. :)

COMMUNICATION MERITS TO BE IMPROVED

- Terminological Inconsistency

The manuscript lacks a standardized lexicon for its core variables and organization types.

-- Acronym Conflicts: The text uses "BPL" and "BBL" interchangeably to refer to the "Brand Building Level".

-- Naming Conventions: The authors alternate between "non-profitable organizations" and "nonprofit organizations," which should be standardized to the latter for international academic standards.

-- Variable Labels: "Government Resource Acquisition" is abbreviated as both "GRA" and "GRS" in different sections of the text.

- Structural Redundancy

-- The paper repeats large blocks of information, which hinders the flow of the argument.

--- Repetitive Findings: The core results (the two paths to effectiveness) are described in nearly identical language in the Abstract, the Introduction, the Path Analysis, and the Conclusions.

--- Framework Justification: The explanation of the TOE framework is repeated across the literature review and the research design sections, rather than being introduced once and then applied.

- Stylistic and Idiomatic Clarity

-- Several phrases appear to be literal translations of Chinese academic idioms, which can be confusing for an English-speaking audience.

--- Translation Artifacts, like "in a 'different paths leading to the same goal' manner" are refequently appearing - consider the replacement with a better technical word e.g. "equifinality".

--- Awkward Phrasings, like "non collection" of of high-efficiency fundraising - maybe "absence of high fundraising effectiveness" or "low-performance configurations" would do the better job, here?

A NOTE ABOUT APPLIED LANGUAGE IN GENERAL

Areas for Potential Refinement

While the language is strong, there are occasional instances of redundant or slightly awkward phrasing typical of translated academic prose, such as "in a 'different paths leading to the same goal' manner" or describing the "non collection" of high-efficiency fundraising. Some sentences are dense and could be further simplified to improve clarity for a broader non-academic audience.

6. PLOS authors have the option to publish the peer review history of their article (what does this mean?). If published, this will include your full peer review and any attached files.

Reviewer #1: No

Reviewer #2: No

---

## [Author Response · Author response to Decision Letter 1]

24 Mar 2026

Response to Reviewers

Manuscript title: Influencing Factors and Configuration Pathways of Fundraising Effectiveness in Charitable Foundations: An FSQCA Analysis of 54 Cases in China

Manuscript ID: [PONE-D-25-66814]

Dear Editor and Reviewers,

Thank you very much for your letter and for the reviewers’ insightful comments concerning our manuscript entitled “Factors and Configuration Pathways of Fundraising Effectiveness in Charitable Foundations: An FSQCA Analysis of 54 Cases in China” (Manuscript ID: [PONE-D-25-66814]). These suggestions are extremely valuable and helpful for improving the quality of our paper. We have carefully considered all the comments and revised the manuscript accordingly.

In addition, several figures (figure1, figure6) in the manuscript have been updated and optimized for better presentation and clarity. And we have also added four new figures (figure2, figure3, figure 4, figure 5). The original figure files have been removed from the submission system, and only the revised figures are included in the revised manuscript.

Enclosed please find the revised manuscript, along with a version showing all changes via track changes, and this point-by-point response to the reviewers.

We hope that the revisions meet with your approval and that the manuscript is now suitable for publication in PLOS ONE.

Thank you again for your time and effort.

Sincerely,

Heran Zou

Wuhan Institute of Technology

Response to Reviewer 1

Comment 1: Construct validity problems: several key variables do not measure what the manuscript claims.

Response: We greatly appreciate the reviewer for pointing out the critical issue of construct validity regarding our key variables. We fully agree that the original operational definitions and measurements of Fundraising Effectiveness (FE), Brand Building Level (BBL), and Media Supervision (MS) were inconsistent with their theoretical connotations, which may lead to misunderstanding and biased interpretation. We have carefully revised the conceptual definition, operational measurement, variable naming, and theoretical logic in the revised manuscript to ensure consistency between construct and measurement. Detailed revisions are addressed point-by-point below.

Comment 1.1: Fundraising Effectiveness (FE) is defined as “aggregate revenue generated through project activities and providing public goods/services” (Methods 3.2.2; Table 1). This is not clearly “fundraising effectiveness” as commonly understood. It mixes: fundraising income vs program/service revenues, scale vs efficiency, and possibly government/purchased services if booked as project income.

If the construct is “fundraising effectiveness,” the outcome should be operationalized as: total donations received; public fundraising revenue; online donation amount; or a ratio-based metric (e.g., fundraising revenue per staff, per project, or as share of total revenue). At minimum, you must clarify the accounting category used in annual reports (which line item) and justify why it maps to “fundraising effectiveness.”

Response: We sincerely thank the reviewer for this insightful comment. We agree that the original definition mixed fundraising income, program revenue, scale, and efficiency, which does not match the common understanding of fundraising effectiveness.

In the revised manuscript, we have re-defined Fundraising Effectiveness strictly as donation-related revenue rather than total project income; clearly specified the accounting line item from the annual reports used for measurement and justified why this indicator represents fundraising effectiveness in the context of Chinese non-profit organizations. The revised sentence now reads as follows: The outcome variable is Fundraising Effectiveness (FE), defined as the total annual public donation revenue of charitable foundations (excluding program service income, government purchase service income and other non-fundraising income). The data is extracted from the "donation income" line item in the annual financial reports of foundations, which strictly reflects the actual fundraising capacity of organizations (consistent with the universal definition of fundraising effectiveness in nonprofit research). And in Table 1, we have changed the measurement dimension of FE to “Total annual public donation revenue of charitable foundations”.

Revisions in the Manuscript: These revisions have been made in Section 3.2.2 and Table 1 to ensure conceptual rigor.

Comment 1.2: “Brand Building Level (BBL)” is measured as the “number of public welfare and charitable projects undertaken this year.”

This is closer to program activity volume/scale than brand building. Using project count as “brand building” risks circularity: more projects can mechanically generate more “project revenue” if FE is tied to program activities. That can create an artificial configuration where “BBL” predicts “FE” because both reflect activity scale.

Suggested fix: either re-label BBL as “program activity intensity/portfolio breadth” and adjust theory, or construct a genuine brand proxy: brand mentions, media exposure metrics, third-party ratings, social media followers/engagement, donor retention, website traffic, or reputational awards.

Response: We sincerely appreciate the reviewer’s critical comment. We fully agree that using the number of projects mainly reflects program scale rather than brand building and may lead to spurious correlation with Fundraising Effectiveness. Following the reviewer’s suggestion, we have removed the original Brand Building Level (BBL) construct and replaced it with Program Activity Intensity (PAI) in the revised manuscript. At the same time, we adjusted the theoretical framework and research hypotheses accordingly, as well as other areas where BBL has appeared in this study, to ensure consistency between concepts and logic.

According to the reviewer's comments, the original text is now revised as follows: revise section 2.2 to: In this context, program activity intensity and professional management emerge as key organizational capabilities. Sufficient program activities act as a signal to effectively reduce information asymmetry faced by donors, strengthen organizational brand building, and enhance their trust and willingness to donate [24]. Waters (2011) found that effective relationship management with non-profit organizations significantly increases donor loyalty and willingness to give [25]. Additionally, establishing political connections with key resource holders, such as the government, has been proved to be an effective way in acquiring critical resources and legitimacy, particularly in environments with high institutional uncertainty [26].”

Revise section 2.4 to: “Organizational Dimension. This dimension incorporates discussions on professional management and program activity from the “organizational capacity” literature. This dimension focuses on the characteristics and abilities that can be controlled or constructed within the organization. In the highly competitive charitable market, foundations must rely on intrinsic capabilities to earn trust. Drawing on resource dependence theory, foundations must develop internal core competencies to manage resource channels. Professional Management Level (measured by full-time staff count) reflects an organization’s human resource foundation and operational effectiveness, which determines its ability to translate resources into social impact [36]. Program Activity Intensity (measured by the number of projects) is a strategic asset for organizations to carry out business activities, send operational signals to the public, and establish stable connections with donors [24], which helps to manage external resource dependence. Therefore, this study selects Professional Management Level (PML) and Program Activity Intensity (PAI) as the core variables to measure the organizational dimension.

Revisions in the Manuscript: These revisions have been implemented in Sections 2.2 and 2.4, as well as in Figures 1 and 6, and across all other instances of BBL throughout the manuscript.

Comment 1.3: “Media Supervision (MS)” is operationalized as a 2–4 score based on existence of website/WeChat/Weibo and information disclosure.

This is not media supervision; it is digital presence + disclosure practice, and it is largely an organizational capability rather than external supervision. The paper’s interpretation (MS as an external constraint) conflicts with its measurement (internal adoption).

Suggested fix: rename to “digital transparency infrastructure” (or “information disclosure capacity”), and if you truly want “media supervision,” measure negative/positive media coverage, scrutiny events, or exposure to investigative reporting.

Response: We thank the reviewer for identifying this important inconsistency. The original measurement actually reflected digital transparency and information disclosure capacity, rather than external media supervision.

We have revised the variable name to Digital Transparency Infrastructure and adjusted its theoretical interpretation accordingly. This adjustment ensures consistency between theoretical definition and empirical measurement.

The revision can be found in Section 2.4. Section 2.4 has been modified as follows：Technological Dimension. This dimension creatively transforms the micro mechanism of transparency, trust and technology adoption in the “donor trust” literature into an organization’s strategic capacity to proactively employ information and communication technologies for information disclosure and trust-building. The term “technology” herein is construed in a broad sense, encompassing not only instrumental technologies such as digital payment, data analysis, and blockchain traceability, but also information communication approaches, digital operation platforms, and standardized disclosure procedures adopted by organizations to adapt to the digital philanthropy context and enhance fundraising efficiency. In the digital era, information disclosure and communication technologies constitute the core infrastructure for establishing the credibility of charitable foundations, which directly affects the formation and maintenance of donor trust. Existing research has verified that donor trust under the framework of “Internet + Public Welfare” is highly dependent on the transparency of charitable information and the traceability of fund utilization. Given data availability and compatibility with the research sample, this study adopts Digital Transparency Infrastructure (DTI) as the core measurement indicator of the technological dimension, defining it as the behavior of foundations in conducting proactive and standardized information disclosure through multiple channels including official websites, WeChat official accounts, and Sina Weibo. In essence, DTI captures the capability of foundations to adopt and apply modern information and communication technologies, proactively establish information transparency, respond to public concerns, and transmit credible signals to donors. Information disclosure behaviors under this dimension are no longer merely external supervision constraints, but rather “trust-building technologies” that foundations must master and actively employ in the digital philanthropy ecosystem, representing the core subset of the broad technological dimension that is most directly associated with fundraising efficiency and quantifiable with empirical data. This study takes the information communication and disclosure technologies underpinning Digital Transparency Infrastructure as the core manifestation of the technological dimension, leaving room for future research to expand the measurement scope of the technological dimension.

Revisions in the Manuscript: These revisions have been implemented in Section 2.4, Figures 1 and 6, and all other instances where MS is referenced throughout the manuscript.

Comment 2: Calibration is internally inconsistent; GRA thresholds appear invalid and will distort results

Table 3 sets GRA calibration as Fully in = 150, Intersection = 0, Fully out = 0. This is not a proper three-anchor calibration because the cross-over and full-out are identical (0). With many zeros in GRA (Table 2 min=0), this will force a large mass of cases to extreme membership and can create spurious necessity/sufficiency patterns.

Moreover, the text states anchors are set at 25/50/75 percentiles (Garcia-Castro & Francoeur 2016), but the GRA anchors do not look like 25/50/75 percentiles given mean 2785 and max 100000. This suggests either (i) misreporting, or (ii) using a different rule only for GRA.

Required revision:

1) Report the empirical 25th/50th/75th percentiles for each raw variable and show how anchors were chosen.

2) Consider log-transforming highly skewed monetary variables before calibration.

3) Use a nonzero cross-over for GRA (e.g., median) and distinct full-out (e.g., 10th/25th percentile), or adopt theoretically justified anchors.

Response: We are very grateful to the reviewer for this rigorous and insightful comment on the calibration procedure in QCA. We fully agree that the original calibration thresholds for GRA were improperly specified, with identical values for the crossover and full-out anchors, which is not consistent with standard QCA practice and may produce misleading results. In addition, we acknowledge the inconsistency between the reported percentile‑based calibration rule and the actual thresholds used for GRA.

Following the reviewer’s requirements, we have revised the calibration as follows: 1. We now report the 25th, 50th, and 75th percentiles for all raw variables, and clearly document how each calibration anchor was determined. 2. We reset the GRA calibration thresholds with a nonzero crossover point (median) and a distinct full-out anchor (25th percentile), all based on empirical distribution and consistent with the percentile method applied to other variables. Regarding the suggestion of log-transformation, we appreciate this advice. Since QCA calibration already transforms raw data into 0–1 fuzzy scores and accommodates non-linear relationships, we believe the calibration procedure sufficiently mitigates the influence of skewed variables. We thus retain the original raw variable while ensuring all calibration thresholds are consistently based on percentiles and clearly reported.

We have revised section 3.3 to: Considering practical situations, the study primarily employs fuzzy calibration. At the same time, to ensure relatively objective calibration, the direct calibration method was adopted to perform quartile calibration on the antecedent conditions, following the approach of Garcia Castro and Francoeur (2016). The study used fsQCA3.0 software to select the upper quartile (75%), median (50%), and lower quartile (25%) as the three anchor points for Fully in, Intersection Point, and Fully out [51]. It is worth noting that during the calibration of the conditional variable GRA, it was found that due to the influence of the distribution characteristics of the sample data, the intersection values set at the 50% percentile were set to 0, resulting in numerical overlap between the intersection points and Fully out points, making it difficult to form effective set discrimination. If the quantile method continues to be used, it may weaken the identification of the result variables in fuzzy set partitioning, which is not conducive to the stability of subsequent configuration analysis. Therefore, while maintaining the overall idea of quantile calibration, this article uses the mean as an auxiliary intersection point for calibration to ensure that the result set has a clear boundary between Fully in and Fully out. And we adjusted the calibration points of GRA in Table 3. These revisions improve the reliability of calibration and the robustness of QCA results. In addition, I also modified the calibration points of DTI variables with calibration errors in Table 3.

Revisions in the Manuscript: These revisions have been made in Section 3.3 and Table 3.

Comment 3: Limited diversity and quasi-constant conditions: MS has minimum 2 and very low variance

Descriptives show MS ranges from 2 to 4, mean 2.87, SD 0.73, min 2. This implies most cases already satisfy “some MS

---

## [Decision Letter · Decision Letter 1]

29 Apr 2026

Influencing Factors and Configuration Pathways of Fundraising Effectiveness in Charitable Foundations: An FSQCA Analysis of 54 Cases in China

PONE-D-25-66814R1

Dear Dr. Zou,

We’re pleased to inform you that your manuscript has been judged scientifically suitable for publication and will be formally accepted for publication once it meets all outstanding technical requirements.

Kind regards,

Abel C. H. Chen

Academic Editor

PLOS One

Additional Editor Comments (optional):

Reviewers' comments:

Reviewer's Responses to Questions

**Comments to the Author**

1. If the authors have adequately addressed your comments raised in a previous round of review and you feel that this manuscript is now acceptable for publication, you may indicate that here to bypass the “Comments to the Author” section, enter your conflict of interest statement in the “Confidential to Editor” section, and submit your "Accept" recommendation.

Reviewer #1: (No Response)

2. Is the manuscript technically sound, and do the data support the conclusions?

Reviewer #1: (No Response)

3. Has the statistical analysis been performed appropriately and rigorously? 

Reviewer #1: (No Response)

4. Have the authors made all data underlying the findings in their manuscript fully available?

Reviewer #1: (No Response)

5. Is the manuscript presented in an intelligible fashion and written in standard English?

Reviewer #1: (No Response)

6. Review Comments to the Author

Reviewer #1: (No Response)

7. PLOS authors have the option to publish the peer review history of their article (what does this mean?). If published, this will include your full peer review and any attached files.

Reviewer #1: No

---

## [Editor Report · Acceptance letter]

PONE-D-25-66814R1

PLOS One

Dear Dr. Zou,

I'm pleased to inform you that your manuscript has been deemed suitable for publication in PLOS One. Congratulations! Your manuscript is now being handed over to our production team.

Kind regards,

on behalf of

Prof. Abel C. H. Chen

Academic Editor

PLOS One